# COMPOSITION AND DECOMPOSITION OF GANs

## ABSTRACT

In this work, we propose a composition/decomposition framework for adversarially training generative models on composed data - data where each sample can be thought of as being constructed from a fixed number of components. In our framework, samples are generated by sampling components from component generators and feeding these components to a composition function which combines them into a **"composed sample"**. This compositional training approach improves the modularity, extensibility and interpretability of Generative Adversarial Networks (GANs) - providing a principled way to incrementally construct complex models out of simpler component models, and allowing for explicit "division of responsibility" between these components. Using this framework, we define a family of learning tasks and evaluate their feasibility on two datasets in two different data modalities (image and text). Lastly, we derive sufficient conditions such that these compositional generative models are identifiable. Our work provides a principled approach to building on pre-trained generative models or for exploiting the compositional nature of data distributions to train extensible and interpretable models.

## 1 INTRODUCTION

Generative Adversarial Networks (GANs) have proven to be a powerful framework for training generative models that are able to produce realistic samples across a variety of domains, most notably when applied to natural images. However, existing approaches largely attempt to model a data distribution directly and fail to exploit the compositional nature inherent in many data distributions of interest. In this work, we propose a method for training **compositional generative models** using adversarial training, identify several key benefits of compositional training and derive sufficient conditions under which compositional training is identifiable.

This work is motivated by the observation that many data distributions, such as natural images, are compositional in nature - that is, they consist of different components that are combined through some composition process. For example, natural scenes often consist of different objects, composed via some combination of scaling, rotation, occlusion etc. Exploiting this compositional nature of complex data distributions, we demonstrate that one can both incrementally construct models for composed data from component models and learn component models from composed data directly.

In our framework, we are interested in modeling **composed data distributions** - distributions where each sample is constructed from a fixed number of simpler sets of objects. We will refer to these sets of objects as **components**. For example, consider a simplified class of natural images consisting of a foreground object superimposed on a background, the two components in this case would be a set of foreground objects and a set of backgrounds. We explicitly define two functions: **composition** and **decomposition**, as well as a set of **component generators**. Each component generator is responsible for modeling the marginal distribution of a component while the composition function takes a set of component samples and produce a composed sample (see figure 1). We additionally assume that the decomposition function is the inverse operation of the composition function.

We are motivated by the following desiderata of modeling compositional data:

- **Modularity:** Compositional training should provide a principled way to reuse off-the-shelf or pre-trained component models across different tasks, allowing us to build increasingly complex generative models from simpler ones.

- **Interpretability:** Models should allow us to explicitly incorporate prior knowledge about the compositional structure of data, allowing for clear "division of responsibility" between different components.

- **Extensibility:** Once we have learned to decompose data, we should be able to learn component models for previously unseen components directly from composed data.

- **Identifiability:** We should be able to specify sufficient conditions for composition under which composition/decomposition and component models can be learned from composed data.

Within this framework, we first consider four learning tasks (of increasing difficulty) which range from learning only composition or decomposition (assuming the component models are pre-trained) to learning composition, decomposition and all component models jointly.

To illustrate these tasks, we show empirical results on two simple datasets: MNIST digits superimposed on a uniform background and the Yelp Open Dataset (a dataset of Yelp reviews). We show examples of when some of these tasks are ill-posed and derive sufficient conditions under which tasks 1 and 3 are identifiable. Lastly, we demonstrate the concept of modularity and extensibility by showing that component generators can be used to inductively learn other new components in a chain-learning example in section 3.4.

The main contributions of this work are:

1. We define a framework for training compositional generative models adversarially.

2. Using this framework, we define different tasks corresponding to varying levels of prior knowledge and pre-training. We show results for these tasks on two different datasets from two different data modalities, demonstrating the lack of identifiability for some tasks and feasibility for others.

3. We derive sufficient conditions under which our compositional models are identifiable.

## 1.1 RELATED WORK

Our work is related to the task of disentangling representations of data and the discovery of independent factors of variations (Bengio et al. (2013)). Examples of such work include: 1) methods for evaluation of the level of disentaglement (Eastwood & Williams (2018)), 2) new losses that promote disentaglement (Ridgeway & Mozer), 3) extensions of architectures that ensure disentanglement (Kim & Mnih (2018)). Such approaches are complementary to our work but differ in that we explicitly decompose the structure of the generative network into independent building blocks that can be split off and reused through composition. We do not consider decomposition to be a good way to obtain disentangled representations, due to the complete decoupling of the generators. Rather we believe that decomposition of complex generative model into component generators, provides a source of building blocks for model construction. Component generators obtained by our method trained to have disentangled representations could yield interpretable and reusable components, however, we have not explored this avenue of research in this work.

Extracting GANs from corrupted measurements has been explored by Bora et al. (2018). We note that the noise models described in that paper can be seen as generated by a component generator under our framework. Consequently, our identifiability results generalize recovery results in that paper. Recent work by Azadi et al. (2018) is focused on image composition and fits neatly in the framework presented here. Along similar lines, work such as Johnson et al. (2018), utilizes a monolithic architecture which translates text into objects composed into a scene. In contrast, our work is aimed at deconstructing the monolithic architectures into component generators.

## 2 METHODS

### 2.1 DEFINITION OF FRAMEWORK

Our framework consists of three main moving pieces:

| Method | Learn components | Learn composition | Learn decom-position | Generative model |
|---|---|---|---|---|
| LR-GAN(Yang et al., 2017) | Background | True | False | True |
| C-GAN (Azadi et al., 2018) | False | True | True | False |
| ST-GAN (Zhang et al., 2017) | False | True | False | False |
| InfoGAN (Chen et al., 2016) | False | False | False | True |

Table 1: Various GAN methods can learn some, but not all, parts of our framework. These parts may exist implicitly in each of the models, but their extraction is non-trivial.

**Component generators** $g_i(\mathbf{z}_i)$ A component generator is a standard generative model. In this paper, we adopt the convention that the component generators are functions that maps some noise vector $\mathbf{z}$ sampled from standard normal distribution to a component sample. We assume there are $m$ component generators, from $g_1$ to $g_m$. Let $\mathbf{o}_i := g_i(\mathbf{z}_i)$ be the output for component generator $i$.

**Composition function** ($c : (\mathbb{R}^n)^m \to \mathbb{R}^n$) Function which composes $m$ inputs of dimension $n$ to a single output (composed sample).

**Decomposition function** ($d : \mathbb{R}^n \to (\mathbb{R}^n)^m$) Function which decomposes one input of dimension $n$ to $m$ outputs (components). We denote the $i$-th output of the decomposition function by $d(\cdot)_i$.

Without loss of generality we will assume that the composed sample has the same dimensions as each of its components.

Together, these pieces define a "composite generator" which generates a composed sample by two steps:

- Generating component samples $\mathbf{o}_1, \mathbf{o}_2, ..., \mathbf{o}_m$.

- Composing these component samples using $c$ to form a composed sample.

The composition and/or decomposition function are parameterized as neural networks.

Below we describe two applications of this framework to the domain of images and text respectively.

## 2.2 EXAMPLE 1: IMAGE WITH FOREGROUND OBJECT(S) ON A BACKGROUND

In this setting, we assume that each image consists of one or more foreground object over a background. In this case, $m \geq 2$, one component generator is responsible for generating the background, and other component generators generate individual foreground objects.

An example is shown in figure 1. In this case the foreground object is a single MNIST digit and the composition function takes a uniform background and overlays the digit over the background. The decomposition function takes a composed image and returns both the foreground digit and the background with the digit removed.

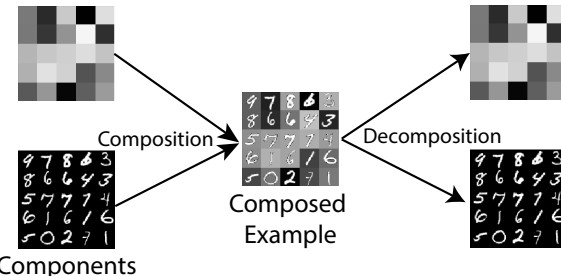

Figure 1: An example of composition and decomposition for example 1.

## 2.3 EXAMPLE 2: COHERENT SENTENCE PAIRS

In this setting, we consider the set of adjacent sentence pairs extracted from a larger text. In this case, each component generator generates a sentence and the composition function combines two sentences and edits them to form a coherent pair. The decomposition function splits a pair into individual sentences (see figure 2).

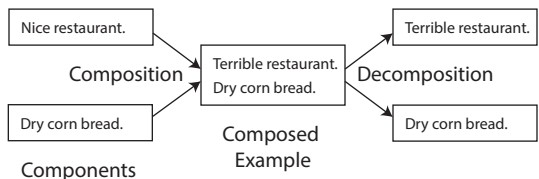

Figure 2: An example of composition and decomposition for example 2.

## 2.4 LOSS FUNCTION

In this section, we describe details of our training procedure. For convenience of training, we implement a composition of Wasserstein GANs introduced in Arjovsky et al. (2017) ) but all theoretical results also hold for standard adversarial training losses.

**Notation** We define the data terms used in the loss function. Let $\mathbf{x}_1$ be a component sample. There are $m$ such samples. Let $\mathbf{y}$ be a composite sample be obtained by composition of components $\mathbf{x}_1, \mathbf{x}_2, ..., \mathbf{x}_m$. For compactness, we use $\mathbf{o}_i$ as an abbreviation for $\mathbf{g}_i(\mathbf{z}_i)$. We denote vector $L^1$ norm by $\|\mathbf{a}\|_1$ ($\|\mathbf{a}\|_1 = \sum_i |a_i|$ ). Finally, we use capital $D$ to denote discriminators involved in different losses.

**Component Generator Adversarial Loss ($l_{\mathbf{g_i}}$)** Given the component data, we can train component generator ($\mathbf{g}_i$) to match the component data distribution using loss

$$l_{\mathbf{g_i}} \equiv \mathbb{E}_{\mathbf{x}_i \sim p_{\text{data}}(\mathbf{x}_i)}[D_i(\mathbf{x}_i)] - \mathbb{E}_{\mathbf{z}_i \sim p_{\mathbf{z}}}[D_i(\mathbf{g_i}(\mathbf{z}_i))].$$

**Composition Adversarial Loss ($l_{\mathbf{c}}$)** Given the component generators and composite data, we can train a composition network such that generated composite samples match the composite data distribution using loss

$$l_{\mathbf{c}} \equiv \mathbb{E}_{\mathbf{y} \sim p_{\text{data}}(\mathbf{y})}[D_c(\mathbf{y})] - \mathbb{E}_{\mathbf{z}_1 \sim p_{\mathbf{z}_1}, ..., \mathbf{z}_m \sim p_{\mathbf{z}_m}}[D_c(\mathbf{c}(\mathbf{o}_1, ..., \mathbf{o}_m))]$$

**Decomposition Adversarial Loss ($l_{\mathbf{d}}$)** Given the component and composite distributions, we can train a decomposition function $\mathbf{d}$ such that distribution of decomposed of composite samples matches the component distributions using loss

$$l_{\mathbf{d}} \equiv \mathbb{E}_{\mathbf{z}_1 \sim p_{\mathbf{z}_1}, ..., \mathbf{z}_m \sim p_{\mathbf{z}_m}}[D_f(\mathbf{o}_1, ..., \mathbf{o}_m)] - \mathbb{E}_{\mathbf{y} \sim p_{\text{data}}(\mathbf{y})}[D_f(\mathbf{d}(\mathbf{y}))].$$

**Composition/Decomposition Cycle Losses ($l_{\mathbf{c-cyc}}, l_{\mathbf{d-cyc}}$)** Additionally, we include a cyclic consistency loss (Zhu et al. (2017)) to encourage composition and decomposition functions to be inverses of each other.

$$l_{\mathbf{c-cyc}} \equiv \mathbb{E}_{\mathbf{z}_1 \sim p_{\mathbf{z}_1}, ..., \mathbf{z}_m \sim p_{\mathbf{z}_m}} \left[ \sum_i \|\mathbf{d}(\mathbf{c}(\mathbf{o}_1, ..., \mathbf{o}_m))_i - \mathbf{o}_i\|_1 \right]$$

$$l_{\mathbf{d-cyc}} \equiv \mathbb{E}_{\mathbf{y} \sim p_{\text{data}}(\mathbf{y})} \left[ \|\mathbf{c}(\mathbf{d}(\mathbf{y})) - \mathbf{y}\|_1 \right]$$

Table 2 summarizes all the losses. Training of discriminators ($D_i, D_c, D_f$ ) is achieved by maximization of their respective losses.

## 2.5 PROTOTYPICAL TASKS AND CORRESPONDING LOSSES

Under the composition/decomposition framework, we focus on a set of prototypical tasks which involve composite data.

Table 2: Table for all losses

| Loss name | Detail |
|---|---|
| $l_{\mathbf{g_i}}$ | $\mathbb{E}_{\mathbf{x}_i \sim p_{\text{data}}(\mathbf{x}_i)}[D_i(\mathbf{x}_i)] - \mathbb{E}_{\mathbf{z}_i \sim p_{\mathbf{z}}}[D_i(\mathbf{g_i}(\mathbf{z}_i))]$ |
| $l_{\mathbf{c}}$ | $\mathbb{E}_{\mathbf{y} \sim p_{\text{data}}(\mathbf{y})}[D_c(\mathbf{y})] - \mathbb{E}_{\mathbf{z}_1 \sim p_{\mathbf{z}_1}, \ldots, \mathbf{z}_m \sim p_{\mathbf{z}_m}}[D_c(\mathbf{c}(\mathbf{o}_1, \ldots, \mathbf{o}_m))]$ |
| $l_{\mathbf{d}}$ | $\mathbb{E}_{\mathbf{z}_1 \sim p_{\mathbf{z}_1}, \ldots, \mathbf{z}_m \sim p_{\mathbf{z}_m}}[D_f(\mathbf{o}_1, \ldots, \mathbf{o}_m)] - \mathbb{E}_{\mathbf{y} \sim p_{\text{data}}(\mathbf{y})}[D_f(\mathbf{d}(\mathbf{y}))]$ |
| $l_{\mathbf{c-cyc}}$ | $\mathbb{E}_{\mathbf{z}_1 \sim p_{\mathbf{z}_1}, \ldots, \mathbf{z}_m \sim p_{\mathbf{z}_m}}\left[\sum_i \|\mathbf{d}(\mathbf{c}(\mathbf{o}_1, \ldots, \mathbf{o}_m))_i - \mathbf{o}_i\|_1\right]$ |
| $l_{\mathbf{d-cyc}}$ | $\mathbb{E}_{\mathbf{y} \sim p_{\text{data}}(\mathbf{y})}\left[\|\mathbf{c}(\mathbf{d}(\mathbf{y})) - \mathbf{y}\|_1\right]$ |

**Task 1:** Given component generators $\mathbf{g_i}, i \in \{1, \ldots, m\}$ and $\mathbf{c}$, train $\mathbf{d}$.

**Task 2:** Given component generators $\mathbf{g_i}, i \in \{1, \ldots, m\}$, train $\mathbf{d}$ and $\mathbf{c}$.

**Task 3:** Given component generators $\mathbf{g_i}, i \in \{1, \ldots, m-1\}$ and $\mathbf{c}$, train $\mathbf{g_m}$ and $\mathbf{d}$.

**Task 4:** Given $\mathbf{c}$, train all $\mathbf{g_i}, i \in \{1, \ldots, m\}$ and $\mathbf{d}$

To train generator(s) in these tasks, we minimize relevant losses:

$$l_{\mathbf{c}} + l_{\mathbf{d}} + \alpha(l_{\mathbf{c-cyc}} + l_{\mathbf{d-cyc}}),$$

where $\alpha \geq 0$ controls the importance of consistency. While the loss function is the same for the tasks, the parameters to be optimized are different. In each task, only the parameters of the trained networks are optimized.

To train discriminator(s), a regularization is applied. For brevity, we do not show the regularization term (see Petzka et al. (2017)) used in our experiments.

The tasks listed above increase in difficulty. We will show the capacity of our framework as we progress through the tasks.

Theoretical results in Section 4 provide sufficient conditions under which Task 1. and Task 3. are tractable.

## 3 EXPERIMENTS

### 3.1 DATASETS

We conduct experiments on three datasets:

1. **MNIST-MB** MNIST digits LeCun & Cortes (2010) are superimposed on a monochromic one-color-channel background (value ranged from 0-200) (figure 3). The image size is 28 x 28.

2. **MNIST-BB** MNIST digit are rotated and scaled to fit a box of size 32 x 32 placed on a monochrome background of size 64 x 64. The box is positioned in one of the four possible locations (top-right, top-left, bottom-right, bottom-left), with rotation between $(-\pi/6, \pi/6)$ (figure 4).

3. **Yelp-reveiws** We derive data from Yelp Open Dataset Yelp Inc. (2004-2018). From each review, we take the first two sentences of the review. We filtered out reviews containing sentences shorter than 5 and longer than 10 words. By design, the sentence pairs have the same topic and sentiment. We refer to this quality as coherence. Incoherent sentences have either different topic or different sentiment.

### 3.2 NETWORK ARCHITECTURES

**MNIST-MB, MNIST-BB** The component generators are DCGAN (Radford et al. (2015)) models. Decomposition is implemented as a U-net (Ronneberger et al. (2015)) model. The inputs to the composition network are concatenated channel-wise. Similarly, when doing decomposition, the

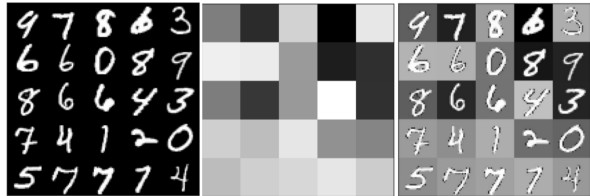

Figure 3: Examples of MNIST-MB dataset. 5x5 grid on the left shows examples of MNIST digits (first component), middle grid shows examples of monochromatic backgrounds (second component), grid on the right shows examples of composite images.

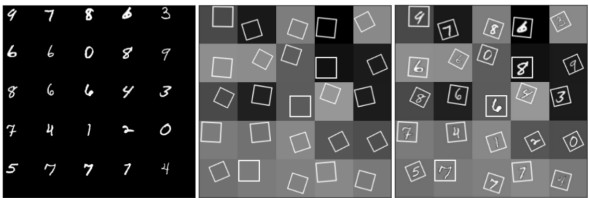

Figure 4: Examples of MNIST-BB dataset. 5x5 grid on the left shows examples of MNIST digits (first component), middle grid shows examples of monochromatic backgrounds with shifted, rotated, and scaled boxes (second component), grid on the right shows examples of composite images with digits transformed to fit into appropriate box.

outputs of the decomposition network are concatenated channel-wise before they are fed to the discriminator.

**Yelp-reviews** The component (sentence) generator samples from a marginal distribution of Yelp review sentences. Composition network is a one-layer Seq2Seq model with attention Luong et al. (2015). Input to composition network is a concatenation of two sentences separated by a delimiter token. Following the setting of Seq-GAN Yu et al. (2017), the discriminator ($D_c$) network is a convolutional network for sequence data.

### 3.3 EXPERIMENTS ON MNIST-MB

Throughout this section we assume that composition operation is known and given by

$$c(\mathbf{o}_1, \mathbf{o}_2)_i = \begin{cases} o_{1,i} & \text{if } o_{2,i} = 0 \\ o_{2,i} & \text{otherwise.} \end{cases}$$

In tasks where one or more generators are given, the generators have been independently trained using corresponding adversarial loss $l_{\mathbf{g_i}}$.

**Task 1:** Given $\mathbf{g_i}, i \in \{1, 2\}$ and $c$, train $\mathbf{d}$. This is the simplest task in the framework. The decomposition network learns to decompose the digits and backgrounds correctly (figure 5) given $c$ and pre-trained generative models for both digits and background components.

**Task 2:** Given $\mathbf{g_1}$ and $\mathbf{g_2}$ train $\mathbf{d}$ and $\mathbf{c}$. Here we learn composition and decomposition jointly 6. We find that the model learns to decompose digits accurately; interestingly however, we note that backgrounds from decomposition network are inverted in intensity ($t(b) = 255 - b$) and that the model learns to undo this inversion in the composition function ($t(t(b)) = b$) so that cyclic consistency ($\mathbf{d}(\mathbf{c}(\mathbf{o}_1, \mathbf{o}_2)) \approx [\mathbf{o}_1, \mathbf{o}_2]$) and $\mathbf{c}(\mathbf{d}(\mathbf{y})) \approx \mathbf{y}$ is satisfied. We note that this is an interesting case where symmetries in component distributions results in the model learning component distributions only up to a phase flip.

**Task 3:** Given $\mathbf{g_1}$ and $\mathbf{c}$, train $\mathbf{g_2}$ and $\mathbf{d}$. Given digit generator and composition network, we train decomposition network and background generator (figure 7). We see that decomposition network learns to generate nearly uniform backgrounds, and the decomposition network learns to inpaint.

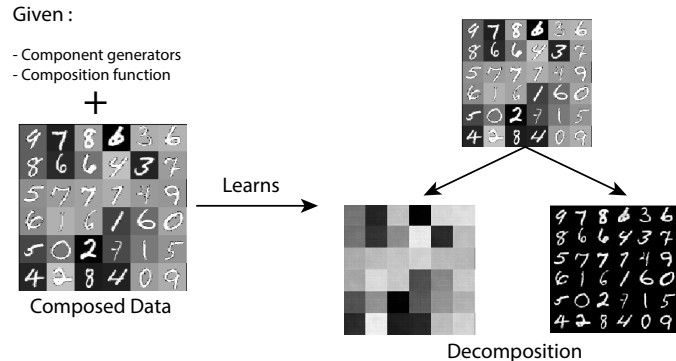

Figure 5: Given component generators and composite data, decomposition can be learned.

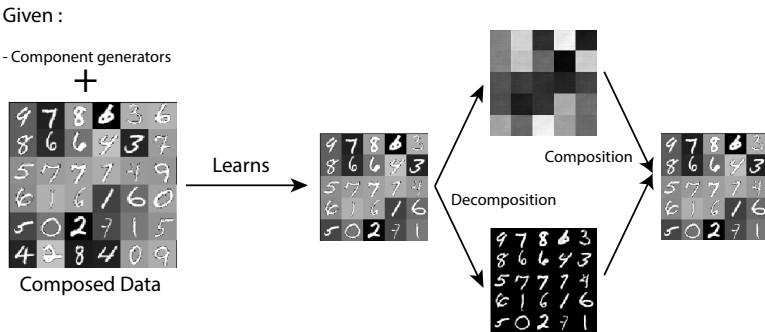

Figure 6: Training composition and decomposition jointly can lead to "incorrect" decompositions that still satisfy cyclic consistency. Results from the composition and decomposition network. We note that decomposition network produces inverted background (compare decomposed backgrounds to original), and composition network inverts input backgrounds during composition (see backgrounds in re-composed image). Consequently decomposition and composition perform inverse operations, but do not correspond to the way the data was generated.

**FID evaluation** In Table 3 we illustrate performance of learned generators trained using the setting of Task 3, compared to baseline monolithic models which are not amenable to decomposition. As a complement to digits we also show results on Fashion-MNIST overlaid on uniform backgrounds (Xiao et al., 2017) (see appendix for examples of task 3 and chain learning on Fashion-MNIST)

**Task 4:** Given $c$, train $g_1, g_2$ and $d$. Given just composition, learn components and decomposition. We show that for a simple composition function, there are many ways to assign responsibilities to different components. Some are trivial, for example the whole composite image is generated by a single component (see figure 11 in Appendix).

### 3.4 Chain Learning - Experiments on MNIST-BB

In task 3 above, we demonstrated on the MNIST-MB dataset that we can learn to model the background component and the decomposition function from composed data assuming we are given a model for the foreground component and a composition network. This suggests the natural follow-up question: if we have a new dataset consisting of a previously unseen class of foreground objects on the same distribution of backgrounds, can we then use this background model we've learned to learn a new foreground model?

We call this concept **"chain learning"**, since training proceeds sequentially and relies on the model trained in the previous stage. To make this concrete, consider this proof-of-concept chain (using the MNIST-BB dataset):

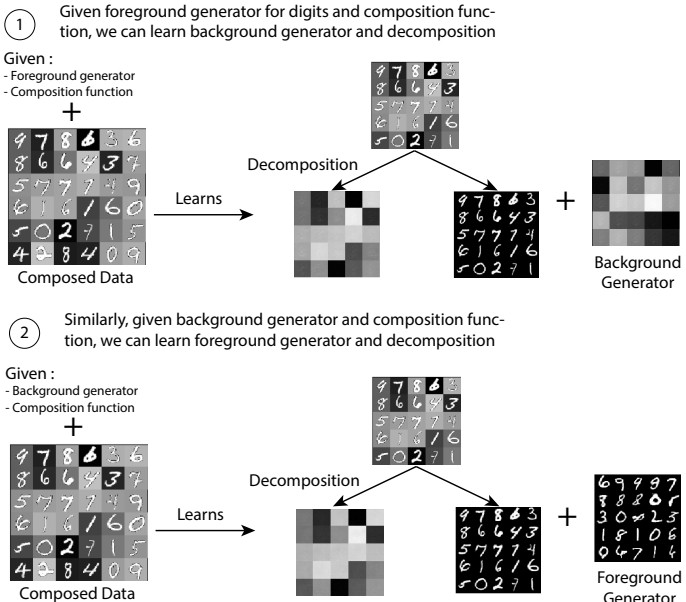

Figure 7: Given one component, decomposition function and the other component can be learned.

0. Train a model for the digit "1" (or obtain a pre-trained model).
1. Using the model for digit "1" from step 1 (and a composition network), learn the decomposition network and background generator from composed examples of "1"s.
2. Using the background model and decomposition network from step 2, learn a model for digit "2" from from composed examples of "2"s.

As shown in figure 8 we are able to learn both the background generator (in step 1) and the foreground generator for "2" (in step 2) correctly. More generally, the ability to learn a component model from composed data (given models for all other components) allows one to incrementally learn new component models directly from composed data.

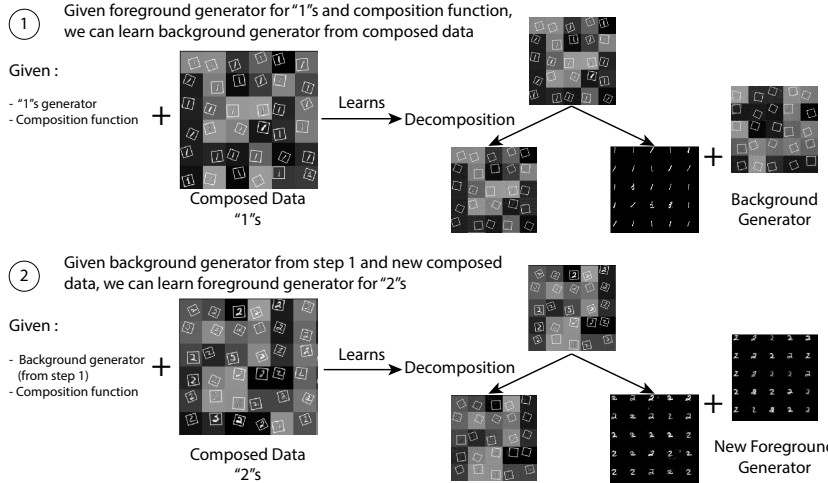

Figure 8: Some results of chain learning on MNIST-BB. First we learn a background generator given foreground generator for "1" and composition network, and later we learn the foreground generator for digit "2" given background generator and composition network.

| Methods | Foreground | | Foreground+background | |
|---|---|---|---|---|
| | Digits | Fashion | Digits | Fashion |
| WGAN-GP | $6.622 \pm 0.116$ | $20.425 \pm 0.130$ | $25.871 \pm 0.182$ | $21.914 \pm 0.261$ |
| By decomposition | $9.741 \pm 0.144$ | $21.865 \pm 0.228$ | $13.536 \pm 0.130$ | $21.527 \pm 0.071$ |

Table 3: We show Frechet inception distance (Heusel et al., 2017) for generators trained using different datasets and methods. The "Foreground" column and "Foreground+background" reflect performance of trained generators on generating corresponding images. WGAN-GP is trained on foreground and composed images. Generators evalueaged in the "By decomposition" row are obtained as described in Task 3 – on composed images, given background generator and composition operator. The information processing inequality guarantees that the resulting generator cannot beat the WGAN-GP on clean foreground data. However, the composed images are better modeled using the learned foreground generator and known composition and background generator.

Example of coherent sentence composition

| Inputs | the spa was amazing ! the owner cut my hair |
|---|---|
| Baseline | the spa was amazing ! the owner cut my hair . |
| $l_{\mathbf{c}}$ | the spa was clean and professional and professional . our server was friendly and helpful . |
| $l_{\mathbf{d-cyc}} + l_{\mathbf{c}}$ | the spa itself is very beautiful . the owner is very knowledgeable and patient . |

Failure modes for $l_{\mathbf{d-cyc}} + l_{\mathbf{c}}$

| Inputs | green curry level 10 was perfect . the owner responded right away to our yelp inquiry . |
|---|---|
| $l_{\mathbf{d-cyc}} + l_{\mathbf{c}}$ | the food is amazing ! the owner is very friendly and helpful . |
| Inputs | best tacos in las vegas ! everyone enjoyed their meals . |
| $l_{\mathbf{d-cyc}} + l_{\mathbf{c}}$ | the best buffet in las vegas . everyone really enjoyed the food and service are amazing . |

Figure 9: Composition network can be trained to edit for coherence. Only the model trained using loss $l_{\mathbf{d-cyc}} + l_{\mathbf{c}}$ achieves coherence in this example. For this model, we show illustrative failure modes. In the first example, the network removes specificity of the sentences to make them coherent. In the second case, the topic is changed to a common case and the second sentence is embellished.

## 3.5 EXPERIMENTS ON YELP DATA

For this dataset, we focus on a variant of **task 1**: given $\mathbf{d}$ and $\mathbf{g_1}, \mathbf{g_2}$, train $\mathbf{c}$. In this task, the decomposition function is simple – it splits concatenated sentences without modification. Since we are not learning decomposition, $l_{\mathbf{c-cyc}}$ is not applicable in this task. In contrast to MNIST task, where composition is simple and decomposition non-trivial, in this setting, the situation is reversed. Other parts of the optimization function are the same as section 2.4.

We follow the state-of-the-art approaches in training generative models for sequence data. We briefly outline relevant aspects of the training regime.

As in Seq-GAN, we also pre-train the composition networks. The data for pre-training consist of two pairs of sentences. The output pair is a coherent pair from a single Yelp review. Each of the input sentences is independently sampled from a set of nearest neighbors of the corresponding output sentences. Following Guu et al. (2017) we use Jaccard distance to find nearest neighbor sentences. As we sample a pair independently, the input sentences are not generally coherent but the coherence can be achieved with a small number of changes.

Discrimination in Seq-GAN is performed on an embedding of a sentence. For the purposes of training an embedding, we initialize with GloVe word embedding Pennington et al. (2014). During adversarial training, we follow regime of Xu et al. (2017) by freezing parameters of the encoder of the composition networks, the word projection layer (from hidden state to word distribution), and the word embedding matrix, and only update the decoder parameters.

To enable the gradient to back-propagate from the discriminator to the generator, we applied the Gumbel-softmax straight-through estimator from Jang et al. (2016). We exponentially decay the temperature with each iteration. Figure 9 shows an example of coherent composition and two failure modes for the trained composition network.

## 4 IDENTIFIABILITY RESULTS

In the experimental section, we highlighted tasks which suffer from identifiability problems. Here we state sufficient conditions for identifiability of different parts of our framework. Due to space constraints, we refer the reader to the appendix for the relevant proofs. For simplicity, we consider the output of a generator network as a random variable and do away with explicit reference to generators. Specifically, we use random variables $X$ and $Y$ to refer to component random variables, and $Z$ to a composite random variable. Let range$(\cdot)$ denote range of a random variable. We define indicator function, $\mathbb{1}[a]$ is 1 if $a$ is true and 0 otherwise.

**Definition 1.** *A* **resolving matrix***, $R$, for a composition function $c$ and random variable $X$, is a matrix of size $|range(Z)| \times |range(Y)|$ with entries $R_{z,y} = \sum_{x \in range(X)} p(X = x)\mathbb{1}[z = c(x,y)]$ (see figure 10).*

**Definition 2.** *A composition function $c$ is bijective if it is surjective and there exists a decomposition function $d$ such that*

*1. $d(c(x,y)) = x, y; \forall x \in range(X), y \in range(Y)$*

*2. $c(d(z)) = z; \forall z \in range(Z)$*

*equivalently, $c$ is bijective when $c(x,y) = c(x',y')$ iff $x = x'$ and $y = y'$. We refer to decomposition function $d$ as* **inverse** *of $c$.*

In the following results, we use assumptions:

**Assumption 1.** *$X, Y$ are finite discrete random variables.*

**Assumption 2.** *For variables $X$ and $Y$, and composition function $c$, let random variable $Z$ be distributed according to*

$$p(Z = z) = \sum_x \sum_y p(Y = y)p(X = x)\mathbb{1}[z = c(x,y)]. \tag{1}$$

**Theorem 1.** *Let Assumptions 1 and 2 hold. Further, assume that resolving matrix of $X$ and $c$ has full column-rank. If optimum of*

$$\inf_{p(Y')} \sup_{\|D\|_L \leq 1} \mathbb{E}_Z[D(Z)] - \mathbb{E}_{X,Y'}[D(c(X,Y'))] \tag{2}$$

*is achieved for some random variable $Y'$, then $Y$ and $Y'$ have the same distribution.*

**Theorem 2.** *Let Assumptions 1 and 2 hold. Further, assume that $c$ is bijective. If optimum of*

$$\inf_d \mathbb{E}_{X,Y}[\|d(c(X,Y))_x - X\|_1] + \mathbb{E}_Z\left[\|d(c(X,Y))_y - Y\|_1\right] + \mathbb{E}_Z[\|c(d(Z)) - Z\|_1] \tag{3}$$

*is 0 and it is achieved for some $d'$ then $d'$ is equal to inverse of $c$.*

## 5 CONCLUSION

We introduce a framework of generative adversarial network composition and decomposition. In this framework, GANs can be taken apart to extract component GANs and composed together to construct new composite data models. This paradigm allowed us to separate concerns about training different component generators and even incrementally learn new object classes – a concept we deemed chain learning. However, composition and decomposition are not always uniquely defined and hence may not be identifiable from the data. In our experiments we discover settings in which component generators may not be identifiable. We provide theoretical results on sufficient conditions for identifiability of GANs. We hope that this work spurs interest in both practical and theoretical work in GAN decomposability.

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

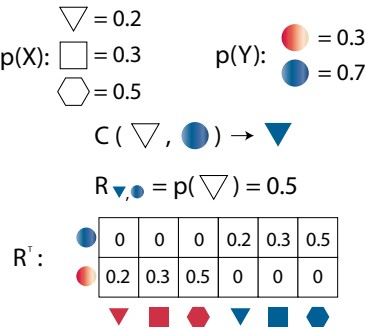

Figure 10: A simple example illustrating a bijective composition and corresponding resolving matrix.

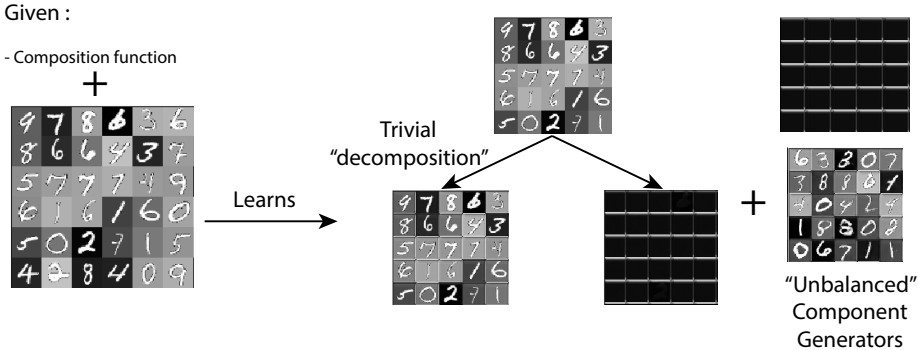

Figure 11: Knowing the composition function is not sufficient to learn components and decomposition. Instead, the model tends to learn a "trivial" decomposition whereby one of the component generators tries to generate the entire composed example.

## 6   IDENTIFIABILITY PROOFS

We prove several results on identifiability of part generators, composition and decomposition functions as defined in the main text. These results take form of assuming that all but one object of interest are given, and the missing object is obtained by optimizing losses specified in the main text.

Let $X, Y$ and $Z$ denote finite three discrete random variables. Let range$(\cdot)$ denote range of a random variable. We refer to $c : \text{range}(X) \times \text{range}(Y) \rightarrow \text{range}(Z)$ as composition function, and $d : \text{range}(Z) \rightarrow \text{range}(X) \times \text{range}(Y)$ as decomposition function. We define indicator function, $\mathbb{1}[a]$ is 1 if $a$ is true and 0 otherwise.

**Lemma 1.** *The resolving matrix of any bijective composition $c$ has full column rank.*

*Proof.* Let $R_{\cdot,y}$ denote a column of $R$. Let $d(z)_x$ denote the $x$ part of $d(z)$, and $d(z)_y$ analogously.

Assume that:

$$\sum_y \alpha_y R_{\cdot,y} = 0 \tag{4}$$

or equivalently, $\forall z \in \text{range}(Z)$:

$$\sum_y \alpha_y \sum_x P(X = x) \mathbb{1}[z = c(x, y)] = 0$$

$$\sum_y \alpha_y \sum_x P(X = x) \mathbb{1}[x = d(z)_x] \mathbb{1}[y = d(z)_y] = 0 \tag{5}$$

$$\alpha_{d(z)_y} P(X = d(z)_x) = 0$$

using the the definition of $R$ in the first equality, making the substitution $\mathbb{1}[z = c(x, y)] = \mathbb{1}[x = d(z)_x] \mathbb{1}[y = d(z)_y]$ implied by the bijectivitiy of $c$ in the second equality and rearranging / simplifying terms for the third.

Since $P(X = x) > 0$ for all $x \in \text{range}(X)$, $\alpha_y = 0$ for all $y \in \{y \mid y = d(z)_y\}$. By the surjectivity of $c$, $\alpha_y = 0$ for all $y \in \text{range}(Y)$, and $R$ has full column rank. □

**Theorem 3.** *Let Assumptions 1 and 2 hold. Further, assume that resolving matrix of $X$ and $c$ has full column-rank. If optimum of*

$$\inf_{p(Y')} \sup_{\|D\|_L \leq 1} \mathbb{E}_Z \left[ D\left( Z \right) \right] - \mathbb{E}_{X,Y'} \left[ D\left( c\left( X, Y' \right) \right) \right] \tag{6}$$

*is achieved for some random variable $Y'$, then $Y$ and $Y'$ have the same distribution.*

*Proof.* Let $Z'$ be distributed according to

$$p(Z' = z) = \sum_x \sum_y p(Y' = y) p(X = x) \mathbb{1}[z = c(x, y)]. \tag{7}$$

The objective in equation 6 can be rewritten as

$$\inf_{p(Y')} \underbrace{\sup_{\|D\|_L \leq 1} \overbrace{\mathbb{E}_Z \left[ D\left( Z \right) \right] - \mathbb{E}_{Z'} \left[ D\left( Z' \right) \right]}^{W(Z,Z')}}_{C(Z')} \tag{8}$$

where dependence of $Z'$ on $Y'$ is implicit.

Following Arjovsky et al. (2017), we note that $W(Z, Z') \to 0$ implies that $p(Z) \xrightarrow{\mathcal{D}} p(Z')$, hence the infimum in equation 8 is achieved for $Z$ distributed as $Z'$. Finally, we observe that $Z'$ and $Z$ are identically distributed if $Y'$ and $Y$ are. Hence, distribution of $Y$ if optimal for equation 6.

Next we show that there is a unique of distribution of $Y'$ for which $Z'$ and $Z$ are identically distributed, by generalizing a proof by Bora et al. (2018) For a random variable $X$ we adopt notation $p_x$ denote a vector of probabilities $p_{x,i} = p(X = i)$. In this notation, equation 1 can be rewritten as

$$p_z = R p_y. \tag{9}$$

Since $R$ is of rank $|\text{range}(Y)|$ then $R^T R$ is of size $|\text{range}(Y)| \times |\text{range}(Y)|$ and non-singular. Consequently, $(R^T R)^{-1} R^T p_z$ is a unique solution of equation 9. Hence, optimum of equation 6 is achieved only $Y'$ which are identically distributed as $Y$. □

**Corollary 1.** *Let Assumptions 1 and 2 hold. Further, assume that $c$ is a bijective. If, an optimum of equation 6 is achieved is achieved for some random variable $Y'$, then $Y$ and $Y'$ have the same distribution.*

*Proof.* Using Lemma 1 and Theorem 3. □

**Theorem 4.** *Let Assumptions 1 and 2 hold. Further, assume that $c$ is bijective. If optimum of*

$$\inf_d \mathbb{E}_{X,Y} \left[ \|d\left( c\left( X, Y \right) \right)_x - X\|_1 \right] + \mathbb{E}_Z \left[ \|d\left( c\left( X, Y \right) \right)_y - Y\|_1 \right] + \mathbb{E}_Z \left[ \|c(d(Z)) - Z\|_1 \right] \tag{10}$$

*is 0 and it is achieved for some $d'$ then $d'$ is equal to inverse of $c$.*

*Proof.* We note that for a given distribution, expectation of a non-negative function – such as norm – can only be zero if the function is zero on the whole support of the distribution.

Assume that optimum of 0 is achieved but $d'$ is not equal to inverse of $c$, denoted as $d^*$. Hence, there exists a $z'$ such $(x', y') = d'(z') \neq d^*(z') = (x^*, y^*)$. By optimality of $d'$, $c(d'(z')) = z'$ or the objective would be positive. Hence, $c(x', y') = z'$. By Definition 2, $c(d^*(z')) = z'$, hence $c(x^*, y^*) = z'$. However, $d'(c(x^*, y^*)) \neq (x^*, y^*)$ and expectation in equation 10 over $X$ or $Y$ would be positive. Consequently, the objective would be positive, violating assumption of optimum of 0. Hence, inverse of $c$ is the only function which achieves optimum 0 in equation 10.

$\square$

## 7 IMPLEMENTATION DETAILS

### 7.1 ARCHITECTURE FOR MNIST / FASHION-MNIST

We use U-Net (Ronneberger et al., 2015) architecture for MNIST-BB for decomposition and composition networks. The input into U-Net is of size 28x28 (28x28x2) the outputs are of size 28x28x2 (28x28) for decomposition (composition). In these networks filters are of size 5x5 in the deconvolution and convolution layers. The convolution layers are of size 32, 64, 128, 256 and deconvolution layers are of size 256, 128, 64, 32. We use leaky rectifier linear units with alpha of 0.2. We use sigmoidal units in the final output layer.

For MNIST-MB and Fashion-MNIST composition networks, we used 2 layer convolutional neural net with 3x3 filter. For decomposition network on these datasets, we used fully-convolutional network (Long et al., 2015). In this network filters are of size 5x5 in the deconvolution and convolution layers. The convolution layers are of size 32, 64, 128 and deconvolution layers are 128, 64, 32. We use leaky rectifier linear units with alpha of 0.2. We use sigmoidal units in the output layer.

The standard generator and discriminator architecture of DCGAN framework was used for images of 28x28 on MNIST-MB and Fashion-MNIST, and 64x64 on MNIST-MB dataset.

### 7.2 ARCHITECTURE FOR YELP-REVIEWS

We first tokenize the text using the nltk Python package. We keep the 30,000 most frequently occuring tokens and represent the remainder as "unknown". We encode each token into a 300 dimensional word vector using the standard GloVe (Pennington et al., 2014) embedding model.

We use a standard sequence-to-sequence model for composition. The composition network takes as input a pair of concatenated sentences and outputs a modified pair of sentences. We used a encoder-decoder network where the encoder/decoder is a 1-layer gated recurrent unit (GRU) network with a hidden state of size 512. In addition, we implemented an attention mechanism as proposed in Luong et al. (2015) in the decoder network.

We adopt the discriminator structure as described in SeqGAN (Yu et al., 2017). We briefly describe the structure at a high level here, please refer to the SeqGAN paper for additional details. SeqGAN takes as input the pre-processed sequence of word embeddings. The discriminator takes the embedded sequence and feeds it through a set of convolution layers of size (200, 400, 400, 400, 400, 200, 200, 200, 200, 200, 320, 320) and of filter size (1, 2, 3, 4, 5, 6, 7, 8, 9, 10, 15, 20). These filters further go through a max pooling layer with an additional "highway network structure" on top of the feature maps to improve performance. Finally the features are fed into a fully-connected layer and produce a real value.

### 7.3 TRAINING

We kept the training procedure consistent across all experiments. During training, we initialized weights as described in He et al. (2015), weights were updated using ADAM (Kingma & Ba, 2014) (with beta1=0.5, and beta2=0.9) with a fixed learning rate of 1e-4 and a mini-batch size of 100. We applied different learning rate for generators/discriminators according to TTUR (Heusel et al., 2017). The learning rate for discriminators is $3 * 10^{-4}$ while for generator is $10^{-4}$. We perform 1 discriminator update per generator update. Results are reported after training for 100000 iterations.

# 8 ADDITIONAL EXAMPLES ON FASHION-MNIST

In this section, we show some examples of learning task 3 (learning 1 component given the other component and composition) as well as an example of cross-domain chain learning (learning the background on MNIST-MB and using that to learn a foreground model for T-shirts from Fashion-MNIST).

As before, given 1 component and the composition operation, we can learn the other component.

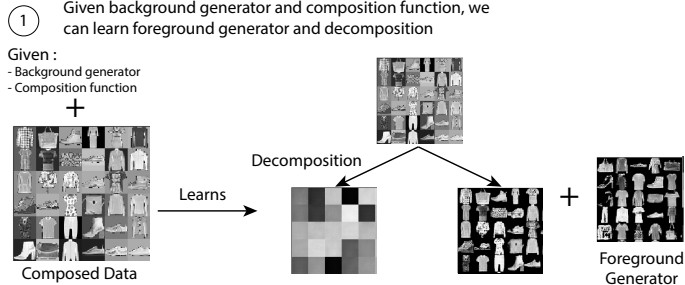

Figure 12: Given one component, decomposition function and the other component can be learned. We show this in the case of Fashion-MNIST

As an example of reusing components, we show that a background generator learned from MNIST-MB can be used to learn a foreground model for T-shirts on a similar dataset of Fashion-MNIST examples overlaid on uniform backgrounds.

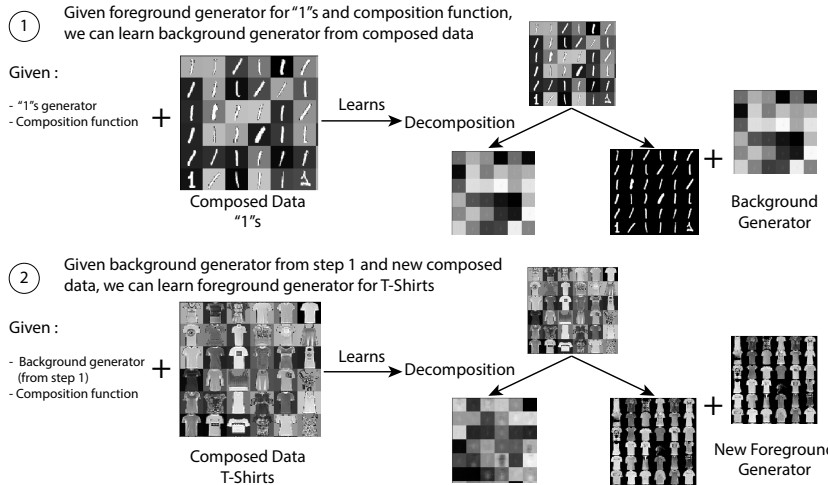

Figure 13: Some results of chain learning on MNIST to Fashion-MNIST. First we learn a background generator given foreground generator for digit "1" and composition network, and later we learn the foreground generator for T-shirts given background generator.

