# OpenReview forum: "COMPOSITION AND DECOMPOSITION OF GANS"
_ICLR.cc/2019/Conference_

### Official Review · AnonReviewer2 · 2018-10-31
**Interesting new problem formulation, not carefully presented and evaluated.**

**Rating:** 4
**Confidence:** 4

**Review:**

The paper proposes a framework for training generative models that work on composed data. The models are trained in an adversarial fashion. The authors apply it to decompose foreground/background parts on MNIST images, and to perform sentence composition/decomposition.

High level comments:
* Clarity: In terms of language and writing style, the paper is written very clearly and easy to follow. In terms of presentation, there are some details that are omitted which would have made understanding easier and the work more reproducible.
* Quality: The idea that is introduced seems intuitive and reasonable, but the experiments does not have enough details to prove that this method works (i.e. no quantitative results presented).  Moreover, the presentation of the method is not very well done (missing details), especially since the authors used the upper limit of 10 pages.
* Originality: I am not familiar with the literature of generative models to judge this precisely, but according to the related work section it sounds like an original idea that is worth sharing.
* Significance: I believe the idea of modeling data composition explicitly sounds intuitive and interesting, and it is worth sharing. However, the experimental section does not have enough evidence that it is actually possible to learn this, so it is not clear whether the contribution is significant.

Pros:
-	interesting new problem formulation
-	simple and clear language
-	the theoretical analysis in the last section could be interesting more generally in the context of GANs
-	the framework is applied on 2 different modalities: images and text.

Cons:
-	hard to tell whether this approach works since the metrics for evaluation are not specified and there are no quantitative results in the experimental section (only 1 qualitative example per task)
-	the work is not reproducible due to the lack of details (see more explanations below)
-	the theoretical analysis is a standalone piece of the paper, without any discussion about the implications, or making connections to the previous sections.

Detailed comments:
1.	I believe the weakest part of this paper is the evaluation section. The authors run their framework on 4 tasks of increasing difficulty. While the MNIST examples make for a nice and intuitive qualitative analysis, the are no quantitative results at all. The only result that is reported for each task is one qualitative picture. The authors make statements such as “The decomposition network learns to decompose the digits and backgrounds correctly” , “Given one component, decomposition function and the other component can be learned.” but there is not mention for how these conclusion are made (no metrics, no numbers). Indeed, it is difficult in general to quantify the results of generative models, but most other GAN papers introduce some sort metric that can be used to aggregate the evaluation on an entire dataset. If the authors manually inspected the results, they should at least report how many images they inspected and how many looked correct.
2.	Aside from evaluation, there are some other details missing from the presentation. The individual details may not be major, but because all of these are missing together, it really affects the overall quality of the paper. For example:
    	 the authors state: “To train discriminator(s), a regularization is applied. For brevity, we do not show the regularization term (see Petzka et al. (2017)) used in our experiments.”. For reproducibility purposes, I believe it is important to at least mention the type of regularization, at least in the appendix.
    	There is a parameter alpha used to balance the losses. What values was used in the experiments?
    	Choices of models are often not explained. Why did you choose that form for c(o1, o2) in section 3.3? Why DCGAN for component generators, and U-net for decomposition?
    	It is not explained in detail how the Yelp-reviews dataset is altered to achieve coherence. The authors mention that “As we sample a pair independently, the input sentences are not generally coherent but the coherence can be achieved with a small number of changes.”. However, the specific algorithm by which these changes are made is not specified, and thus it can’t be reproduced.
3.	The theoretical section is an interesting contribution, but the paper just states a list of theorems without making any connections to the applications used before, or a broader discussion about how these fit in the context of GANs more generally.
4.	My understanding is that both datasets used are created by the authors by making alterations to MNIST and Yelp-reviews dataset, thus making them to some extent synthetic datasets suited to fit this problem formulation. I would have like to see how this composition/decomposition works on existing datasets with no alterations. Does it still work?
5.	In section 2.3, in the coherent sentence experimental setting, I don’t fully understand the design of the task. Figure 2 shows an example where composition and decomposition are not symmetric (i.e. composing then decomposing does not go back to the input sentences), although one of your losses is supposed to ensure exactly this cyclic consistency. Why not choose another problem that doesn’t directly violate your assumptions?

Minor issues:
6.	From the related work section, it is not clear how your approach is different from Azadi et al. (2018). Please include more details.
7.	In section 2.4, you mention using Wasserstein GANs, with no further details about this model (not even a one line description). Without reading their paper, the readers of your paper could not easily follow through this section. The losses further introduced are also not explained intuitively (e.g. what do the two expectation terms in l_g_i represent?).
8.	I believe there are some errors in which tasks reference which figures in section 3.3. Should Task 2 refers to Figure 6, and Task 3 to Figure 7?
9.	What exactly is range(.) in section 4? If this refers to the interval of values that a variable can take, the saying “is a matrix of size |range(Z)| × |range(Y )|” doesn’t exactly make sense. Please define formally.

Final remarks and advice:
Overall, I believe the paper introduces some interesting ideas. There is definitely value in the problem definition and theoretical analysis. However, I believe the paper needs more work on presentation and evaluation, especially since the authors opted for 10 pages and according to ICLR guidelines “Reviewers will be instructed to apply a higher standard to papers in excess of 8 pages.”. Hopefully the above comments will help the authors improve this work!

---

> ### Author Response · Authors · 2018-11-27
> **Thank you reveiwer 2. Please see our reply.**
>
> We thank the reviewer for their detailed and thoughtful review. We have made some improvements to our paper based on these suggestions - adding quantitative evaluations, adding comparisons to relevant related work and clarifying our notation  - please see inline for our detailed responses:
>
> >>> Hard to tell whether this approach works since the metrics for evaluation are not specified… ...
> >>> 1.    I believe the weakest part of this paper is the evaluation section.  ... ...
> (<<<) We agree with the reviewer that while our qualitative results provide some intuition about which tasks are feasible and which are not, providing qualitative metrics across the entire dataset is important. We supplemented our original qualitative results with quantitative metrics - specifically, we evaluated the foreground generator learned from composed examples using FID score and compared this to our base GAN model trained on the actual foreground dataset (as a theoretical upper bound on performance for the compositional model). We show that as expected, we do not do quite as well when we have to learn to decompose and model the foreground simultaneously, but are within range of the FID scores reported in literature on MNIST and Fashion-MNIST.
> >>> 2.    Aside from evaluation, there are some other details missing from the presentation. ... ...Choices of models are often not explained. ...
> (<<<)We apologize for the missing details, these details were omitted due to space constraints but we have included the relevant details on the full architecture used, including type of regularization, values of alpha etc., in a new section of the appendix.
> >>>   It is not explained in detail how the Yelp-reviews dataset is altered to achieve coherence. ... ....
> (<<<) The general architecture of the composition network is described at a high level in section 3.2. In brief, it is a seq-to-seq model that takes the concatenation of the two sentence and outputs a sentence pair that is made more coherent by this network. In addition, we have included additional details on architecture and hyperparameters used in a new section of the appendix.
> >>> 3.    The theoretical section is an interesting contribution, ... ...
> (<<<) We apologize for the disconnected presentation of our theoretical results. The theoretical results were meant to formalize the intuition from the experimental examples that task 1 and 3 are “feasible” in some sense and to provide sufficient conditions on the composition operation such that tasks 1 and 3 are identifiable. We have edited the text to make this connection clearer.
> >>> 4.    My understanding is that both datasets used are created by the authors by making alterations to MNIST and Yelp-reviews dataset, .... ....
> (<<<) Our primary goal was to suggest a set of composition / decomposition subtasks (c.f. tasks 1 through 4 in our submission), as well as deriving some basic theoretical results about the identifiability of these tasks (e.g., conditions where one can learn component models from composed data etc). The experimental results were intended more as illustrative examples of when such models were learnable (or not) which motivated our synthetic datasets where the “ground truth” composition operation is known to us. We agree with the reviewer that it would be interesting to apply our model to more complex datasets and we look forward to exploring that further (along with various extensions of the model that this would require) in future work.
> >>> 5.    In section 2.3, in the coherent sentence experimental setting,... ...
> Minor issues:
> >>> 6.    From the related work section, it is not clear how your approach is different from Azadi et al. (2018). Please include more details.
> (<<<) Complicated but special case of our framework, hence comparison would not be suitable. We cited them
> We have added a comparison table1 which explains how our work relates to various other contributions in this area including Azadi et al.
> >>> 7.    In section 2.4, you mention using Wasserstein GANs, ... ...
> (<<<) We apologize that due to space constraints we were not able to explain the Wasserstein GAN in sufficient detail. We have provided additional details of our architectures in the appendix.
> >>> 8.    I believe there are some errors in which tasks reference which figures in section 3.3. Should Task 2 refers to Figure 6, and Task 3 to Figure 7?
> (<<<) Yes, that is correct, we apologize for the confusion and have corrected the references.
> >>> 9.    What exactly is range(.) in section 4? If this refers to the interval of values that a variable can take, the saying “is a matrix of size |range(Z)| × |range(Y )|” doesn’t exactly make sense. Please define formally.
> (<<<) “range(.)” refers to the set of values that Z and Y can take on “| range (X) |” thus denoting the cardinality of the range of X (the number of values X can take on).

---

### Official Review · AnonReviewer3 · 2018-11-02
**So isolated from the similar works on GANs**

**Rating:** 5
**Confidence:** 5

**Review:**

- There have been works on this before in the GAN literature, they have not been even cited, let alone being compared to in the experiments. Seminal examples include Donahue et al., ICLR 2018 "Semantically decomposing the latent spaces of generative adversarial networks", and (a bit less starkly in terms of the alignment with the goals of this paper): Huang et al., 2017 "Stacked generative adversarial networks".

- In general, comparisons to state-of-the-art (or to other) algorithms are missing.

- Is the assumptions of pre-trained components viable with image, and not text, data? Please elaborate

- The related work section is missing out on dozens of  works, those on disentanglement or interpretability; what is the point then of making a related work section in the first place if only one single example of an algorithm in each broad topic is mentioned? If so, I would suggest mentioning this single example prior to the discussing the topic without a related work section, or (apparently the better option) to do a related work section with a rigorous coverage. Examples of some related works on disentanglement and interpretability:
Higgins et al., ICLR 2017 "beta-VAE" - Kim & Mnih, ICML 2018 "Disentangling by factorising" - Adel et al., ICML 2018 "Discovering interpretable representations for both deep generative and discriminative models" - Chen et al., NIPS 2017 "InfoGAN: Interpretable representation learning by information maximizing generative adversarial nets", etc.

- The advantages promised in Section 1 are a little bit too presumptuous. Too many idealistic assumptions are need in order for these advantages to hold. For instance, extensibility has been mentioned as an advantage in Section 1 and in the abstract, and that has not been capitalised on, or confirmed in the experiments, or from this point onwards.

- It will be interesting to see what happens with rather real-world cases like occlusion, etc

- Writing has room for improvements, in terms of both the flow and also grammar, etc. There are a few typos.

---

> ### Author Response · Authors · 2018-11-27
> **Thank you reveiwer 3. Please see our reply.**
>
> >>> - There have been works on this before in the GAN literature, they have not been even cited, let alone being compared to in the experiments. Seminal examples include Donahue et al., ICLR 2018 "Semantically decomposing the latent spaces of generative adversarial networks", and (a bit less starkly in terms of the alignment with the goals of this paper): Huang et al., 2017 "Stacked generative adversarial networks".
> - In general, comparisons to state-of-the-art (or to other) algorithms are missing.
>
> We thank the reviewer for pointing us to some of the related work in this field. We’ve added a new comparison table that compares our method to other related methods, and in particular, show that to the best of our knowledge we are the first to tackle the general problem of learning a part generator and composition / decomposition directly from composed data. Regarding comparisons specifically to Donahue et al. and Huang et al. please see the section below on “factorized” representations.
>
> >>> - Is the assumptions of pre-trained components viable with image, and not text, data? Please elaborate
>
>
> We apologize for the confusion caused by our presentation of the text example. The assumption of pre-trained components is indeed still viable with text. In our example, the equivalent pre-trained component would be a generative model for sentences from the review corpus - in our experiments, this is done by training a generator on the first/second sentence of reviews in the corpus.
>
> >>> - The related work section is missing out on dozens of  works, those on disentanglement or interpretability; what is the point then of making a related work section in the first place if only one single example of an algorithm in each broad topic is mentioned? If so, I would suggest mentioning this single example prior to the discussing the topic without a related work section, or (apparently the better option) to do a related work section with a rigorous coverage. Examples of some related works on disentanglement and interpretability:
> Higgins et al., ICLR 2017 "beta-VAE" - Kim & Mnih, ICML 2018 "Disentangling by factorising" - Adel et al., ICML 2018 "Discovering interpretable representations for both deep generative and discriminative models" - Chen et al., NIPS 2017 "InfoGAN: Interpretable representation learning by information maximizing generative adversarial nets", etc.
>
>
> Again, we thank the reviewer for pointing us to work that is related from the interpretability, disentanglement side of things. As the reviewer correctly points out, there is much recent work in the area of learning disentangled representations of data. However, this work is not directly relevant to our work (as we explain below). We included some examples (including Kim & Mnih etc.) from the disentanglement literature, mostly as a means of explaining to the reader how our work differs from the general approaches in learning disentangled representations and not as a comprehensive review of work in the broad field.
>
> As mentioned in our related work section, the work in disentangled representations is complementary to our work but differs in that 1) we specifically attempt to learn standalone component generators and 2) our composition operations occurs “independently” of sampling the components (we also cite this as a reason why the decomposition learned by our models are unlikely to yield good “disentangled” representations). This is important since our goal is to be able to learn marginal, component generators that can then be reused (e.g., to inductively learn more components as illustrated in our chain learning example).
>
> It is unclear to us, for example, how we could sample individual components from a factorized latent representation (nor should that be the goal when optimizing for interpretability). We see our work as a parallel but complementary approach, focusing on exploring how we can build complex models incrementally with atomic generators.
>
> We also thank the reviewer for their feedback on our related work section, we’ve fleshed out comparisons to relevant related work in the form of a new table 1 which summarizes our contribution relative to some of the most similar existing work.
>
> >>> - The advantages promised in Section 1 are a little bit too presumptuous. Too many idealistic assumptions are need in order for these advantages to hold. For instance, extensibility has been mentioned as an advantage in Section 1 and in the abstract, and that has not been capitalised on, or confirmed in the experiments, or from this point onwards.
>
> We agree with the reviewer that the advantages in section 1 are more aspirational and should be reworded to reflect that. We do note that we have added a cross-dataset chain-learning example which we hope does suggest that these advantages could be realizable using this compositional framework.

---

### Official Review · AnonReviewer1 · 2018-11-04
**interesting paper, but missed quantitative analysis and comparisons.**

**Rating:** 4
**Confidence:** 5

**Review:**

[Overview]

In this paper, the authors studied the problem of composition and decomposition of GANs. Motivated by the observations that images are naturally composed of multiple layouts, the authors proposed a new framework to study the compositional image generation and its decomposition by defining several tasks. On those various tasks, the authors demonstrate the possibility of the proposed model to composing image components and decompose the images afterwards. These results are interesting and insightful to some extent.

[Strengthes]

1. The authors proposed a framework for compose images from components and decompose the images into components. Based on this new framework, the authors tried different settings, by fixing the learning of one or more modules in the model. The experiments on various tasks are appreciated.

2. In the experiments, the authors tried both image and text to demonstrate the concepts in this paper. Moreover, some qualitative results are presented.

[Weaknesses]

1. The authors performed multiple experiments regarding various tasks defined in this paper.However, I can hardly find any quantitative evaluation for the results. It is not clear to me that how the quality of the composed images and the decomposed components from images are. I would suggest the authors derive some metric to measure quality quantitatively, provide some statistics on the whole datasets.

2. In this paper, the authors proposed multiple tasks in terms of which parts are fixed and known in the training process. However, dominated by so many different tasks, the core idea is losses in the paper. From the paper, I cannot get the core idea the authors want to deliver. I would suggest the authors focus on one certain task and perform more qualitative and quantitative analysis and comparisons, as also mentioned above.

3. The proposed model has several tricky parts. First, the number of components are pre-determined. However, in realistic cases, the number of components are unknown, and thus how many component generators should be used is ill-posed. Second, the composing operation is simple and tricky. Such a simple composing operation make it hard to adapt to some more complicated data, such as cifar10 or so. Thirdly, almost all tasks need some components known. Even for the Task 4, c is known, and the model performs poorly for generating the disentangled components.

4. The authors missed one very relevant paper:

LR-GAN: Layered Recursive Generative Adversarial Networks for Image Generation. Yang et al.

In the above paper, the authors proposed an end-to-end model for generating images with background and foreground compositionally. It can be applied to a number of realistic datasets. Regardless of the decomposition part in this paper, the proposed method in the above paper seems to be clearly superior to the composition part in this paper considering this paper fails on Task 4. The authors should give credit to the above paper (even the synthesized MNIST dataset looks similar ) and pay some efforts to explain the advantages in comparison it.

[Summary]

This paper proposed a new framework to study the compositionally of images during generation and decomposition. Through several experiments on various tasks, the authors presented some interesting results and provided some insights on the potentials and difficulties in this direction. However, as pointed above, I think this paper lacks enough experimental analysis and comparison. Its core idea hard to capture. Also, it missed a comparison to some related work.

---

> ### Author Response · Authors · 2018-11-27
> **Thank you reveiwer 1. Please see our reply.**
>
> We thank the reviewer for their detailed and thoughtful review. We have made some improvements to our paper based on these suggestions - adding quantitative evaluations and expanding our comparison to related work - please see inline for our detailed responses:
>
> Reply to 1:
>
> First, we’d like to clarify that the primary intent of our work was to suggest a set of composition / decomposition subtasks (c.f. tasks 1 through 4 in our submission), as well as deriving some basic theoretical results about the identifiability of these tasks (e.g., conditions where one can learn component models from composed data etc). The experimental results were intended more as illustrative examples of when such models were learnable (or not) which explained our lack of quantitative evaluations.
>
> However, we agree with the reviewer that providing a qualitative evaluation across the entire dataset is useful. We supplemented our original qualitative results with quantitative metrics - specifically, we evaluated the foreground generator learned from composed examples using a standard FID score and compared this to our base GAN model trained on the actual foreground dataset (as a theoretical upper bound on performance for the compositional model). We show that, as expected, we do not do quite as well when we have to learn to decompose and model the foreground simultaneously, but are within range of the FID scores reported in literature on MNIST. We further evaluated FID scores on Fashion-MNIST in the same manner as an additional validation.
>
> Reply to 2:
>
> We apologise for the lack of clarity in our presentation of the various sub-tasks. Part of the contribution of our work is to enumerate various composition/decomposition tasks and to demonstrate the feasibility of a subset of these tasks. However, we agree with the reviewer that this may result in confusion for the reader. We’ve edited the introduction to make it clearer that our main focus is to demonstrate that “chain learning” is possible since it provides a simple proof-of-concept for modular extensions of GANs.
>
> Reply to 3:
>
> We agree that having a pre-specified number of components is a limitation of this framework. We are definitely interested in exploring extensions of such models beyond a fixed, pre-specified number of components. However, we believe that even this constrained version of compositionality has not been extensively explored - especially in terms of our theoretical understanding of when such compositional training is possible.
>
> Regarding our compositional operation being too simple, we agree that our composition transformations are not sufficient to capture “real-world” composition. Our goal was to show a proof-of-concept on a challenging but still feasible set of composition operations (e.g., in our chain learning example, the composition consists of scaling, rotation and masking).
> Lastly, we agree that most of the tasks assume knowledge of a component generator. This was the main motivation behind our work (how to re-use GANs in a modular fashion), we believe that the chain learning example shows a possible approach for how one can iteratively build up a collection of component generators and hence handle compositional data of increasing complexity.
>
> Reply to 4:
>
> We thank the reviewer for the pointer to LR-GANs, that is certainly very interesting and relevant related work. However, there are some key differences between our work and the work on LR-GANs. Firstly, we learn a marginal component model for the foreground that is able to generate foreground samples (instead of generating foreground conditioned on background) this is important for us to be able to reuse component generators as demonstrated in our chain learning examples (we have included a cross-domain chain learning example in the appendix to further illustrate this). Secondly, the LR-GAN is restricted to modelling affine compositions and do not learn a corresponding decomposition operation. The authors also demonstrate that both having a good foreground mask and restricting composition to affine transformations is required for good performance of their model in their ablative analysis. We appreciate the insights provided by the authors of LR-GAN, and while these priors are useful when modeling images specifically and may be useful in our contexts as well, we are more focused on identifying where compositional learning is identifiable more generally without
>
> In summary, there are two main differences in the model formulation directly.  First, in our framework the foreground generator and background generator are independent, while LR-GAN’s foreground generator is dependent on background generator. This independence is required for part generators learnt to be reusable. Second, in our framework there is a decomposition operation and a cycle consistency regularization in the model. We showed that this regularization is beneficial to learning a good part generators.

---

### Public Comment · (anonymous) · 2018-11-06
**Not clear how to spatially transform objects**

Dear Authors,

Could you please provide more details on your generators’ architectures? I am particularly interested in your MNIST-BB experiment (Figure 4) and the fact of rotating and shifting digits according to the given background. I think a Resnet or a Unet generator is not able to rotate, shift, and scale objects by an adversarial loss function. This is why a spatial transformer was used in the ST-GAN [1] and Compositional GAN [2] papers. It would be great if you could clarify this.

[1]: Lin, et al. "ST-GAN: Spatial Transformer Generative Adversarial Networks for Image Compositing."
[2]: Azadi, et al. "Compositional GAN: Learning Conditional Image Composition."

---

> ### Author Response · Authors · 2018-11-27
> **Thanks for your comments.**
>
> Dear sir,
>
> Our generators' architectures followed DCGAN.
> A thing to clarify in MNIST-BB experiment is our composition/decomposition network are Unet and they are not generators. For the details about the networks' architectures, please see apendix.
> In terms of your question, Unet composition network shows it can learn rotate, shift and scale(not very large scale) in our experiments. We observed that pooling (in our case is stride=2) is especially important for learning large shifting of foregrounds. We also observed that a fully-conolutional network (encoder-decoder) can achieve similar performance. We actually tried spatial transformer (ST) in our composition network but we failed. The reason is this network needs to learn large shifting that is too far for the gradient to propate for ST. On the other hand, ST-GAN uses a progessive algorithm to update transformation with also support the idea that it is hard to do large warping at one step. Compositionnal GAN uses RAFN network to first change the viewpoint of an object then does spatial transforming that might suggest only ST is not enough. It is not clear whether ST is sufficient to learn all affine transformation under GANs setting.

---

### Meta-Review · Area_Chair1 · 2018-12-14
**Paper falls short of experimental results, especially comparison to state-of-the-art baselines**

**Confidence:** 4
**Recommendation:** Reject

**Metareview:**

This paper investigates composition and decomposition for adversarially training generative models that work on composed data. Components that are sampled from component generators are then fed into a composition function to generate composed samples, aiming to improve modularity, extensibility, and interpretability of GANs. The paper is written very clearly and is easy to follow.
Experiments considered application to both images (MNIST) and text (yelp reviews).
The original version of the paper lacks any qualitative analysis, even though experiments were described. Authors revised the paper to include some experimental results, however, they are still not sufficient. State-of-the-art baselines, from previous work suggested by the reviewers should be included for comparison.